# Harnessing Immune Evasion Strategy of Lymphatic Filariae: A Therapeutic Approach against Inflammatory and Infective Pathology

**DOI:** 10.3390/vaccines10081235

**Published:** 2022-08-01

**Authors:** Priyanka Bhoj, Namdev Togre, Vishal Khatri, Kalyan Goswami

**Affiliations:** 1Independent Researcher, El Paso, TX 79902, USA; priyanka.rhoya@gmail.com; 2Department of Biological Sciences, University of Texas, El Paso, TX 79968, USA; 3Independent Researcher, Woburn, MA 01801, USA; vi.kha9@gmail.com; 4All India Institute of Medical Sciences, Saguna, Kalyani 741245, India

**Keywords:** human lymphatic filariae, immunomodulators, immunomodulation, inflammatory diseases, parasitic diseases

## Abstract

Human lymphatic filariae have evolved numerous immune evasion strategies to secure their long-term survival in a host. These strategies include regulation of pattern recognition receptors, mimicry with host glycans and immune molecules, manipulation of innate and adaptive immune cells, induction of apoptosis in effector immune cells, and neutralization of free radicals. This creates an anti-inflammatory and immunoregulatory milieu in the host: a modified Th2 immune response. Therefore, targeting filarial immunomodulators and manipulating the filariae-driven immune system against the filariae can be a potential therapeutic and prophylactic strategy. Filariae-derived immunosuppression can also be exploited to treat other inflammatory diseases and immunopathologic states of parasitic diseases, such as cerebral malaria, and to prevent leishmaniasis. This paper reviews immunomodulatory mechanisms acquired by these filariae for their own survival and their potential application in the development of novel therapeutic approaches against parasitic and inflammatory diseases. Insight into the intricate network of host immune-parasite interactions would aid in the development of effective immune-therapeutic options for both infectious and immune-pathological diseases.

## 1. Introduction

Lymphatic filariasis is an infection in humans caused by filarial parasites: *Wuchereria bancrofti*, *Brugia malayi*, and *B. timori*. To ensure effective transmission, these parasites evolved with multiple hosts, including a human as a definitive host and the mosquito as an intermediate host. Adult worms can live for up to 7 years in a human, and their microfilarial (Mf) stage can circulate in the bloodstream for up to 9 months. While biting an infected human, mosquitoes ingest microfilariae. Here, microfilariae molt twice and grow into the infective larval (L3) stage. When infected mosquitoes bite another human host, they deposit L3 filariae on the skin, which travel through the cutaneous tissues to the local lymphatics, rapidly develop into the L4 stage, and then migrate to the central lymphatic system, where they develop into sexually mature adults. After mating with male adults, female worms produce approximately 10,000 microfilariae per day, continuing the transmission cycle. A constant co-evolutionary race between the parasite and its host leads to mutual co-existence without causing harm to the host. Their success implies masterful immune evasion strategies. To achieve long-term survival in different anatomical compartments of the host, parasites release a variety of products that are stage- and gender-specific, reflecting specific developmental processes and diverse strategies for evasion of host immune responses [1]. Evaluation of these products will pave the way for a better understanding of how these filarial parasites orchestrate immune evasion.

In 1994, the WHO/United Nations Development Programme/World Bank sponsored the Filarial Genome Project (FGP) to study and map the *B. malayi* genome (the first parasitic nematode that was sequenced) [2]. Comparative genomic and proteomic analyses of *B. malayi* with other parasitic and free-living nematodes have led scientists to understand the genetic and molecular basis of the host–parasite relationship, including the parasite’s immune avoidance tactics [1,3,4]. As a result of these analyses, new potential immunomodulatory and therapeutic targets were identified and prioritized for further in vivo and clinical studies. Many recent reviews have shed light on the immunoregulatory role of helminth-derived fractions and individual products. In this review, we begin with immunomodulatory strategies acquired by human–tropic lymphatic filarial parasites to establish parasitism. Furthermore, we highlight the potential of these filariae and their immunomodulatory molecules in the development of novel therapeutic approaches against various parasitic and inflammatory diseases. A better comprehension of the relationship between the host immune system, filarial infection, and other co-infections can provide new insights into effective immunological treatments.

## 2. Immunomodulation Strategies

Within 3 h of exposure to large and migrating larvae, the host initiates an early acute inflammatory response that elicits a Th1 cytokine response (IFN-γ, TNF-α, IL-1α, IL-8, and granulocyte-macrophage colony-stimulating factor) within 24 h [5,6]. However, this may also cause significant and undesirable tissue damage. Therefore, to maintain homeostasis, the host switches to a different immune profile against these long-lived filariae 7 days post-infection. This immune-regulatory response is of the IL-10-dominated modified Th2-type, characterized by dysfunction of antigen-presenting cells (APC); increased levels of anti-inflammatory cytokines (IL-4, IL-10, and TGF-β), regulatory T cells, and alternatively activated macrophages (AAM); and induction of immune cell apoptosis. Unlike Th1- or Th17-type immune responses induced by other microbial infections, the modified Th2-type response controls inflammation, repairs tissues injured during infection and migration, and restores homeostasis in an infected host [7].

Filariae manipulate the host’s defense system by producing and releasing a variety of bioactive chemicals and extracellular vesicles (helminth-derived particles (HDPs)) that attack the host’s intracellular and extracellular immune apparatus. Certainly, comprehensive efforts to characterize the full immunomodulatory abilities of filariae and their HDPs are still in their infancy, making this a large and prospective area of research.

### 2.1. Infective L3 Mute Cutaneous Innate Cells

In response to microbial infection, damaged cells produce alarmins such as thymic stromal lymphopoietin (TSLP), IL-25, and IL-33. These cytokines activate type 2 innate lymphoid cells (ILC2), a major contributor of anti-helminthic immunity that triggers and amplifies type 2 inflammation [8]. However, L3 filariae masterfully evade the immune response in the cutaneous tissue. Although the presence of ILC2 and alarmins, IL-25 and IL-33, has not been studied directly at the site of infection (skin), previous studies have demonstrated that L3 filariae do not induce ILC2 stimulating cytokines IL-18 and TSLP [9,10]. The spatial difference between ILCs (found mainly in the upper dermal layer) and Langerhans cells (LCs, found mainly in the epidermal layer) in the skin [11] can be one reason for the failure of cutaneous ILC activation. LCs and dermal dendritic cells (DCs) remain quiescent [10] and fail to initiate an ILC2-dependent inflammatory response to L3 filariae. Filarial molecules responsible for muting this cutaneous immune response remain unexplored and warrant further investigation. Strikingly, *W. bancrofti* infected Mf+ patients showed increased cKit^+^ ILCs in the peripheral blood that drives Th17 immune response [12], which may be involved in immune-pathologic consequences [5,13]. However, significant increases in IL-4 alone or in both IL-4 and IL-10 producing CD4^+^ cells, IL-10-producing adaptive T regulatory cells (aTreg/Tr1), and natural T regulatory cells (nTregs) in Mf+ patients at homeostasis are likely to downregulate the immune response [13]. Overall, this suggests a heterogeneous ILC response to different filarial stages and anatomical locations in the host.

### 2.2. Filariae Regulate PRR Signaling

Innate immune cells are equipped with pattern recognition receptors (PRRs) that recognize specific pathogen-associated molecular patterns (PAMPs) and trigger intracellular downstream pathways that evoke pro-inflammatory responses. Most PRRs can be classified into Toll-like receptors (TLRs), nucleotide oligomerization domain (NOD)-like receptors (NLRs), retinoic acid-inducible gene-I (RIG-I)-like receptors (RLRs), C-type lectin receptors (CLRs), and absent in melanoma-2 (AIM2)-like receptors (ALRs) [14].

During immunomodulation, mostly glycan conjugated products (glycoproteins and glycolipids) target TLRs and CLRs. This strategy of parasitic survival, called “glycan gimmickry”, not only alters TLR expression, but also masterly manipulates its intracellular signaling [15,16]. For instance, *B. malayi* and *W. bancrofti* secrete glycoproteins, such as leucyl aminopeptidase [17,18,19], a homolog of a phosphorylcholine (PC)-carrying glycoprotein ES-62 of *Acanthocheilonema viteae* [20]. In monocytes, ES-62 forms a complex with TLR4. Subsequent internalization of ES-62–TLR4 complexes drives caveolae/lipid raft-dependent sequestration and autophagolysosomal degradation of protein kinase C-α, a molecule essential for mast cell activation [21] (Figure 1A). In DCs, ES-62 downregulates TLR4-associated protein kinase C-δ (PKC-δ), upregulates and sequesters p62 and LC3 (components of an autophagy machinery), induces their autophagolysosomal degradation, and suppresses the LPS-driven release of IL-6, IL-12p70, and TNF-α [22] (Figure 1B). ES-62-induced degradation of PKC-δ is supposed to hamper DC development and motility, IL-12p40/p70 expression, MHC II-Ag presentation, and Th1 polarization. Although the effect of purified *B. malayi* homologs of ES-62 has not been studied yet, *B. malayi* microfilariae were found to significantly downregulate the expression of TLR4, downstream molecules (MyD88 protein and the binding ability of p50 and p65), and pro-inflammatory IL-12p40 in LPS activated DCs [23]. In contrast, phosphorylcholine-binding *W. bancrofti* sheath antigen induces DC maturation and pro-inflammatory cytokine secretion via the TLR4-dependent pathway that drives Th1 and regulatory T cell responses [24]. TLR4-dependent downstream signaling in DC subsets with differing cellular responses invites additional mechanistic research regarding the role of these glycans in immune response modulation. *Setaria cervi* thioredoxin reductase (TrxR) shows 100% sequence identity with *B. malayi* TrxR isoform C. In addition to its antioxidant activity, TrxR possesses anti-inflammatory activity in macrophages with inhibition of the TLR4/NF-κB axis, down-regulation of the inflammasome pathway, and activation of AAM [25] (Figure 1C).

### 2.3. HDPs Mimic Host Glycans and Immune Molecules

Another approach used by parasites is molecular mimicry, in which filariae express host-like glycan antigens on their surface to evade recognition by the host’s immune system. Ludin et al. (2011) provided a list of molecular mimicry candidates from human parasites, including *B. malayi* [26]. Microfilarial and adult stages of *B. malayi* express TGH-2, a human TGF-β homolog, and activate immunosuppressive mechanisms through TGF-βR signaling [27,28]. Another example is the *B. malayi* protein BmHsp12.6, which has an IL-10-like function in addition to chaperone activities [29]. Microfilariae of *W. bancrofti* and *B. malayi* secrete prostaglandins, mainly PGE2 [30]. PGE2 plays a diverse role in immune regulation [31], including induction of FOXP3^+^ Treg cells [32,33], vasodilation, and inhibition of platelet aggregation [30,34]. Similar to human macrophage migration inhibitory factor (MIF), *B. malayi* MIF-1 and -2 activate human monocytes to produce pro-inflammatory cytokines IL-8, TNF-α, and endogenous MIF in vitro [35]. However, in the presence of IL-4, *B. malayi* MIF promotes AAM differentiation, implying that Bm-MIF plays a different role on macrophages depending on the prevailing cytokine environment [36]. Moreover, the sex specific fucosylation of MIF-1 is believed to enhance its immune-suppressive function, demanding further investigation of its glycosylated state [37].

### 2.4. Filariae Manipulate B and T Cell Response

Among several immune pathways, the effect of filarial parasites on Tregs with concomitant secretion of immunosuppressive cytokines (IL-10 and TGF-β) is a primary mechanism of controlling inflammation (Figure 2). For instance, *W. bancrofti* sheath antigen induces DC activation through TLR4 signaling, followed by Th1 and Treg cell elicitation [24]. Strikingly, in another study, a *W. bancrofti*-infected population showed distinct IL-10-producing regulatory B and T cell subsets, which is helpful for the parasite’s survival [38]. At the time of infection, *B. malayi* larvae secrete abundant larval transcript (Bm-ALT) protein, which is associated with upregulation of GATA-3 transcription factor in macrophages, inducing a Th2 immune response. Moreover, Bm-ALT induces SOCS-1, inhibits IFN-γR associated JAK kinase in macrophages, and interferes with signals required for the development of pro-inflammatory Th1 cells [39]. BmK1 protein from *B. malayi* selectively blocks voltage-gated potassium (Kv) 1.3 channels and suppresses IFN-γ production in CCR7^-^ effector memory T cells, but not in naïve or central memory T cells [40].

Researchers have shown that PC alone, PC-BSA, and PC carrying ES-62 directly inhibit polyclonal activation of B cells in a PKC-dependent manner, implying that PC-containing molecules in *B. malayi* can imitate immunosuppressive action [41]. Notably, PC is not identified on the ES-62 homolog, leucyl aminopeptidase (LAP), in *B. malayi*, but on another secretory protein called N-acetylglucosaminyltransferase [42]. It is known that Treg cells and IL-10 induce IgG4 production by B cells. Immunosuppressive IgG4 is neither able to activate the complement system nor induce antibody-dependent cell-mediated cytotoxicity (ADCC) after binding to CD16 on neutrophils and eosinophils [43].

Some filarial molecules exploit the immune response in a receptor-independent manner. For example, apart from receptor-mediated modulation of immune cells [44], *B. malayi* cystatins (BmCys) inhibit host cysteine proteases and asparaginyl endopeptidase, impair antigen presentation on APCs, and reduce T cell priming [45]. Another atypical filarial immunomodulator is a *B. malayi* polyprotein “ladder”, gp15/400, that exploits the immune response in a metabolite-dependent manner. It is suggested that this retinoid binding protein enhances vitamin A uptake by host tissues [46]. In the presence of retinoic acid (a vitamin A metabolite), TGF-β inhibits IL-6-driven TH17 cell proliferation and enhances FOXP3^+^ Treg cell differentiation [47].

### 2.5. Filarial Parasites Induce Immune Cell Apoptosis and Autophagy

Filarial parasites prolong their infection by inducting apoptosis to lower the immune cell population. *B. malayi* L3 filariae activate NK cells to produce IFN-γ and TNF-α, which facilitates cell death via the caspase-dependent pathway [48]. *B. malayi* Mf affect human DCs in two ways: (1) by modifying their function and (2) by triggering cell death, resulting in an antigen-specific T cell hypo-response. Microfilariae interact with human DCs to form cell–parasite aggregates, trigger DC apoptosis in a TRAIL- and TNF-alpha-dependent manner, and impair their ability to produce IL-12, limiting CD4^+^ T cell activation and proliferation [23,49]. Apart from apoptosis, microfilariae trigger autophagy in DCs by inhibiting phosphorylation of mTOR and its downstream proteins (p70S6K1 and 4E-BP1), upregulating Beclin 1 phosphorylation, inducing LC3II, and degrading p62 [50]. Recently, researchers demonstrated that microfilariae release extracellular vesicles that DCs quickly internalize. These extracellular vesicles are rich in unique miRNAs that target the mTOR signaling pathway [51]. *W. bancrofti* has been shown to trigger apoptosis of CD4^+^ T cells via FasL-expressing B-1 cells (induced by elevated IL-10 levels), resulting in a hypo-immune response in infected patients [52].

### 2.6. Non-Cellular Immune Evasion by Filarial Parasites

Filarial parasites are armed with high levels of antioxidant enzymes and non-enzymatic anti-oxidants—glutathione peroxidase, superoxide dismutases, glutathione-s-transferase (GST), thioredoxin peroxidase, TrxR, glutathione (GSH), ascorbic acid, translationally controlled tumor protein, and α-tocopherol—that play an important role in protection against free radicals generated during host immune cell attack [1,46,53,54,55]. In addition, *B. malayi* secretes acetylcholinesterases, which may prevent fluid accumulation in the gut and inhibit parasite clearance [46]. Another *B. malayi* protein, calreticulin, binds to human C1q and blocks further classical complement pathways [56].

These are not the only mechanisms used by filarial parasites to ensure their own survival and prevent extensive damage to the host’s body. A variety of unexplored mechanisms and complex molecules facilitate immune evasion. The evidence for human lymphatic filariae, however, is limited.

## 3. Filarial Immunomodulatory Strategy as a Treatment against Diseases

### 3.1. Lymphatic Filariasis

In 1997, the World Health Assembly set the goal of eliminating lymphatic filariasis globally by 2020 through mass drug administration (MDA). During MDA, all individuals living in endemic areas received one of these single-dose two-drug combinations: albendazole (ALB) + diethylcarbamazine (DEC) citrate; ALB + ivermectin (IVM) in areas co-endemic for onchocerciasis; or ALB, preferably twice a year, in areas co-endemic for loiasis. However, numerous obstacles stand in the way of successful implementation. These drugs are only effective against microfilariae and not against adult and larval parasites. By the end of 2020, MDA had not yet been delivered to ten endemic countries [57], which raised concerns about the recurrence of filarial infections in countries or areas that were previously declared free of LF infection [58]. One reason for this concern is human migration from endemic to LF-free areas [59,60,61,62]. The majority of migrants were from rural endemic areas, which had poor sanitation, rice fields, and inadequate mosquito control. Moreover, climate change and delays in MDA due to COVID-19 are likely to further sabotage eradication efforts [63,64]. According to the WHO’s 2021 report, 859 million people in 50 countries are at risk of lymphatic filariasis, which requires preventive treatment. As a result, the WHO revised the target date to 2030, using a triple-drug MDA combination of IVM, DEC citrate, and ALB (IDA-MDA), which may result in patient non-compliance [65,66,67,68]. This evidence demands the development of effective vaccines and novel therapeutics.

Strikingly, current antifilarial drugs target the immunomodulatory arsenal of filariae; they alter the host-parasite interface, unmasking the host immune system to access the parasite. The widely used drug DEC is believed to block PGI2 and PGE2 production in both microfilariae and endothelial cells. The resulting vasoconstriction enhances endothelial adhesion and microfilariae immobilization as well as destruction by host platelets and granulocytes [69]. IVM prevents protein release from microfilarial extracellular vesicles by blocking the GluCl channel. These proteins are indispensable for evading the host immune system [70]. Maclean et al. (2021) recently investigated the effects of DEC and IVM treatment on the *B. malayi* gene expression that may be responsible for filarial clearance from blood circulation [71]. For example, treatment with either IVM or DEC downregulated galectin expression in adults. Galectins, among many other immunomodulatory effects, impede lymphocyte trafficking [72], stimulate alternative macrophage activation [73], and cause T cell apoptosis [74]. Since oxidative and xenobiotic detoxification mediated by antioxidants is a fundamental survival strategy for filariae, we synthesized and studied the library of sulphonamide chalcones that affect filarial GSH status, produce oxidative stress, and lead to apoptosis [75,76].

Indeed, drugs can heal existing infections, but they will not prevent infections unless they, or their active metabolites, are removed slowly from the host system, and remain in circulation for a lengthy period. Given that filariae orchestrate the host’s immune system for their own growth and survival, manipulating the host’s defense system against LF could be a viable prophylactic option. Several potential vaccine candidates have been identified and tested for their potential against LF [77]. Many antigens are non-homologous to human and immunomodulatory proteins that subvert the host’s immune response against the parasite.

Table 1 summarizes immune-regulatory proteins that have been evaluated as vaccine candidates. *B. malayi* immunomodulatory proteins such as heat shock protein 12.6 (BmHsp12.6αc), abundant larval transcript-2 (Bm-ALT-2), and tetraspanin large extracellular loop (Bm-TSP LEL), showed maximum protection in mouse challenge experiments [29,78,79]. To improve the protective efficacy of monovalent vaccines, these best vaccine candidates were fused to prepare a single multivalent vaccine, rBmHAT (BmHsp12.6 + BmALT-2 + BmTSPLEL). Strikingly, it showed >95% protection against *B. malayi* infection in mice when AL007 or AL019 was used as an adjuvant [80]. However, when administered with alum in non-human primates, rBmHAT provided ~35% protection [81], hinting at a need to change the adjuvant and/or multivalent formulation before using this vaccine in human clinical trials. Adding another immunomodulatory antigen, thioredoxin peroxide (BmTPX-2), to rBmHAT showed >88% protection against the challenge infection [82]. This tetravalent rBmHAXT confers approximately 57% protection against challenge infections in a primate model, which meets the WHO requirement, and hence offers great potential for using this vaccine in human clinical trials [83].

### 3.2. Malaria

Co-infections are common in endemic regions. Control of intracellular pathogens, such as *Plasmodium* species that cause malaria, *Leishmania donovani*, *Mycobacterium tuberculosis* (Mtb), and human immunodeficiency virus (HIV), requires pro-inflammatory Th1 (IL-12, IFN-γ, and TNF-α) and Th17 (IL-17A and IL-23) responses. Increasing evidence suggests that filariae-driven Th2 and Treg immunity can negatively affect the host’s ability to combat these pathogens.

The effect of filarial co-infection on *Plasmodium* spp. has already been discussed in detail [94]. Human and animal studies on LF/malaria co-infection have provided conflicting results, with some demonstrating more severe malaria in the presence of filarial co-infections and others suggesting filariae-induced protection against malaria, depending on the infection severity and parasite type [95,96,97,98,99,100]. A strong Th1 immune response plays a major role in controlling primary malaria infection. However, the filariae-induced IL-10-dependent Th2 immune response modulates inflammatory IL-12p70/ IFN-γ pathways and increases resistance to malaria [98]. Moreover, pre-existing filarial infection can impair the immunogenicity of anti-Plasmodium vaccination, as evidenced by decreases in plasmodium antigen-specific CD8^+^ T cells, IFN-γ, and TNF-α production, resulting in reduced cytotoxicity and protection against malarial infection [101]. A simple solution to filarial interference with vaccination efficacy is deworming before vaccination [102]. However, there are several obstacles to drug-induced abolition of filarial infection in endemic locations. These include (1) the lack of an adulticidal or adult-sterilizing drug or vaccine, (2) the time it takes to return to a normal immune response, and (3) the risk of re-infection during the recovery period. Therefore, it is desirable to optimize appropriate vaccination regimes that elicit a multifaceted and potent immune response in filariae-infected individuals [101,102].

Unlike acute malaria, cerebral malaria and malarial sepsis are triggered due to exaggerated pro-inflammatory responses; filariae-derived immunosuppression can protect against this severe immunopathology [95,99]. However, maintaining a filarial infection to avoid an inflammatory exacerbation is not a smart option. In-depth study is required to strike a delicate balance between permissive filarial infection that does not progress to lymphatic filariasis and appropriate immunosuppression that does not lead to severe complications of malaria. Therapies that imitate filariae-derived immunosuppression may be investigated for the treatment of cerebral malaria.

### 3.3. Leishmaniasis

Leishmaniasis, the third most common vector-borne disease after malaria and lymphatic filariasis, is caused by the protozoan *Leishmania* parasite. Visceral leishmaniasis, also known as kala-azar, is caused by *L. donovani* and *L. infantum* throughout Asia, North Africa, Latin America, and Southern Europe. Every year, 700,000 to 1 million new cases are reported. The WHO actively encourages research into effective leishmaniasis control [57]. Fractions derived from *B. malayi* were found to cross-react with sera from hamsters infected with *L. donovani*, suggesting that these filarial cross-reactive molecules can contribute to the development of anti-leishmanial prophylactics [103]. In vivo studies in hamsters demonstrated that *B. malayi* L3/adult worms or immunization with a fraction of the adult parasite extract (BmAFII) inhibited the progression of both filarial and *L. donovani* infections [104,105]. Recently, studies have shown that heat shock protein 60 (BmHSP60) shares several antigenic regions of B- and T-cell epitopes of leishmania counterparts and protects against leishmanial infection via Th1-mediated immune responses and NO production [103,106]. In contrast, a fraction of *L. donovani* (Ld1) that cross-reacted with sera of *B. malayi* infected animals facilitated filarial infection. Ld1 consists of eight proteins, including HSPs [107]. Therefore, more comprehensive and in-depth investigations are needed to optimize and develop prophylactics based on cross-reactive rationale in co-endemic regions.

The prevalence of filarial and leishmanial co-infections has been reported in some parts of the world [108]. In the co-infected mouse model, local immune responses to filarial and leishmanial infections were polarized and compartmentalized [109]. These findings contradict acute malarial findings in which microfilariae and *Plasmodium* share the same niche—blood. In popliteal lymph nodes (which drain the *L. major* infection site) and thoracic lymph nodes (which drain the *L. sigmodontis* infection site) immune responses were IFN-γ- and IL-4-dominant, respectively. Moreover, pre-existing helminth infection delayed IFN-γ production and *L. major*-induced lesion progression [109]. Notably, unlike the leishmanial co-infection model, which confines parasites to the thoracic cavity, the presence of human lymphatic microfilariae in the bloodstream may provide a different immune outcome. Appropriate filarial animal and human population studies are needed to assess whether the immune response to LF/leishmaniasis co-infection is defensive or progressive; such assessment will aid in the development of appropriate and specific immune modulation therapies.

### 3.4. Inflammatory Diseases

In developed societies, large-scale deworming programs, reduced exposure to infection due to vaccination, and improved sanitation are associated with an increase in the occurrence of inflammatory and metabolic disorders, supporting the hygiene hypothesis [110,111,112]. The ability of parasitic worms to shift the immune response from Th1 to Th2/Treg has sparked interest in employing live worms as immunotherapy. However, rather than reintroducing an infection, one approach to reducing the incidence of inflammatory and autoimmune disorders is to employ substitutes for these infections that retain their protective benefits.

Many studies regarding the therapeutic potential of helminthic proteins in inflammatory diseases have recently been discussed [113,114]. The use of non-human helminthic proteins may be one reason for unsuccessful clinical trials. Human filariae have co-evolved with the human immune system, suggesting that it is more suitable to use human filarial proteins over other helminthic proteins. There is strong evidence in mouse models that human filarial therapy, excretory-secretory components, and their recombinant molecules can treat and/or prevent inflammatory diseases such as inflammatory bowel disease (IBD), type-1-diabetes (T1D), and rheumatoid arthritis (RA) (Table 2). However, potential filarial proteins have only been tested in the laboratory and have not been tested in clinical trials. Effective coordination can reduce duplication of work, as many proteins have the same mode of action in different inflammatory diseases. For example, rBmALT-2 has been found to reduce the severity of T1D and IBD by downregulating IFN-γ and upregulating IL-10 and IgG1/IgG2a [115,116]. Although promising results have been achieved with human lymphatic filarial therapy, many questions, such as those regarding optimal dose, treatment duration, immunization route, safety profile, and cellular mode of action, remain unanswered. There is considerable scope for research in this area. For instance, site-directed administration of filarial immunomodulatory proteins using anti-colitic probiotics can provide effective IBD prevention/cure therapy [117]. A series of research studies, ranging from basic to clinical, is essential to evaluate the efficacy, safety, tolerability, and ethical implications of genetically modified immunobiotics.

## 4. Conclusions and Future Prospects

A millennia-long co-evolutionary struggle between the filarial parasite and its host has resulted in a mutually advantageous coexistence; the parasite’s sophisticated immune evasion strategies do not perceptibly harm the host. Filariae manipulate the host’s intracellular and extracellular machinery to create an immunomodulatory milieu to suppress inflammation by producing and releasing various bioactive chemicals and extracellular vesicles. Future research should delve into the involvement of filarial proteins with unknown functions in filariae’s intriguing regulatory effects on the host’s immune system.

Manipulation of the filariae-induced anti-inflammatory immune response can be a rational treatment approach not only against filariae, but also against intracellular parasites such as *Plasmodium* spp. and *L. donovani* in LF co-infected regions. The presence of LF in co-endemic regions either negatively affects the inflammatory protective response against microbes or positively prevents immunopathologies produced by an exacerbated host response to chronic microbial infection. Therefore, the contribution of filarial infection should be considered when conducting vaccine trials against non-helminthic infections in co-endemic areas. Compared with existing targeted medicines, helminths have proven to be safer and more controllable immunotherapies for acute and chronic inflammatory diseases due to their ability to activate immunoregulatory circuits and reduce inflammation. However, to date no helminth therapy has demonstrated promising results in human trials, suggesting the need for more in-depth work to identify and test more specific molecules as therapeutic candidates.

Indeed, comprehensive efforts to characterize human filarial immunomodulators against inflammatory diseases are still in their infancy; however, their significant translational potential must be explored. The emerging immunosuppressive role of filarial parasites in sepsis, inflammaging, and organ transplantation warrants further investigation. As filarial parasites and tumors both influence the host’s immune response through immunosuppression and immunological blinding, a multidisciplinary approach will advance research by encouraging crosstalk among diverse fields of science. Finally, deciphering the mechanisms of immune-modulation by filarial proteins provides information that can be used to design new strategies, not only for the treatment of filariasis and other infections, but also for the treatment of immunopathological diseases.

## Figures and Tables

**Figure 1 vaccines-10-01235-f001:**
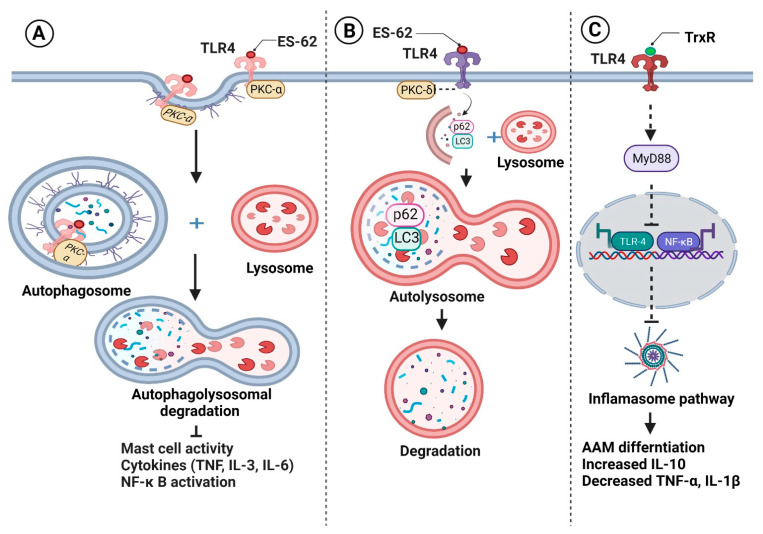
Filariae regulate pattern recognition receptors signaling for their survival. Filarial glycoproteins such as ES-62 (**A**) form complexes with Toll-like receptor 4 (TLR) on monocytes, followed by their internalization and sequestration with protein kinase C-α and (**B**) down-regulate TLR4-associated protein kinase C-δ (PKC-δ) in dendritic cells, followed by sequestration of p62 and LC3. As a result, sequestered components are degraded by autophagolysosomes, inhibiting the production of pro-inflammatory cytokines and inflammatory cell polarization. (**C**) Thioredoxin reductase (TrxR) inhibits the TLR4/NF-κB pathway, prevents inflammasome activation, and induces alternatively activated macrophages (AAM) activation, which is characterized by an increase in anti-inflammatory IL-10 synthesis and a decrease in pro-inflammatory cytokines. Figure created using Biorender.com.

**Figure 2 vaccines-10-01235-f002:**
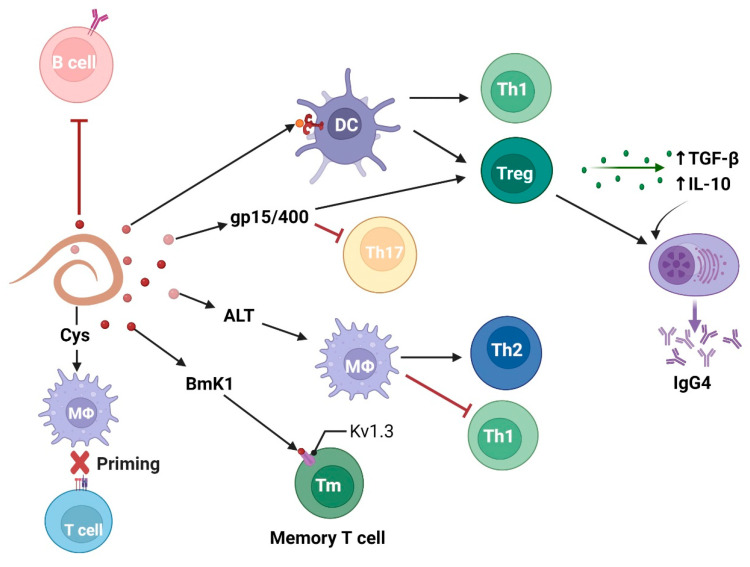
Filarial parasites modulate adaptive immune cell response characterized by T regulatory and anti-inflammatory Th2 cells with concomitant release of immunosuppressive cytokines (IL-10 and TGF-β). Figure created using Biorender.com.

**Table 1 vaccines-10-01235-t001:** Filarial immune-regulatory proteins that have been evaluated as vaccine candidates.

Candidate Vaccine	Outcome	Reference
Serpin (BmSPN-2)	Immune response is strong but short-lived, suggesting that serpins alone are not effective vaccine candidates for long-term immunity	[84]
Aabundant larval transcript-2 (BmALT-2)	BmALT-2 protein and Bm-alt-2 DNA conferred approximately 75% and 57% protection, respectively	[78]
Glutathione-S-transferases (WbGST)	61% protection in Jirds challenge experiments and 65.5% protection in in situ challenge studies	[29,85,86]
Small heat shock protein HSP12.6 (BmHsp12.6αc subunit)	83% protection in mouse in situ challenge studies	
Large extracellular loop of tetraspanin (BmTSP-LEL)	64% protection in mouse in situ challenge studies	[79]
Bivalent vaccines: HSP12.6 + ALT-2, HSP12.6 + TSP-LEL, TSP-LEL + ALT-2	90%, 80%, and 82% protection in mouse in situ challenge studies, respectively	[87]
Trehalose-6-phosphate phosphatase (BmTPP)	78.4% decrease in microfilariae counts and 71% reduction in adult parasite load in Mastomys	[88]
Thioredoxin (WbTRX), Thioredoxin peroxidase (WbTPX)	57% and 62% protection in Mastomys challenge experiments, respectively	[89]
BmHAT Trivalent vaccine	Protein and DNA protein prime boost vaccination yielded approximately 95% protection in mice 3 out of 5 vaccinated macaques were protected from challenge infection	[81,87]
Cystatin-2 in which the amino acid Asn66 was mutated to Lys66 (Bm-CPI-2M)	48.6% and 48.0% at 42 and 90 days post-infection, respectively, with *B. malayi* L3 filariae	[90]
BmALT-2 with Tuftsin as fusion protein	65% larvicidal activity in ADCC experiments	[91]
Calreticulin (BmCRT)	Offers protection during experimental lymphatic filariasis	[92]
BmHAXT Tetravalent vaccine	88% protection in mouse in situ challenge studies 57% protection in rhesus macaques challenge infections Reduced fecundity and adult worm burden in Jirds	[82,83,93]

Abbreviations: BmHAT refers to BmHsp12.6 + BmALT-2 + BmTSPLEL; and BmHAXT refers to BmHsp12.6 + BmALT-2 + BmTSPLEL + BmTPX-2.

**Table 2 vaccines-10-01235-t002:** Human lymphatic filariae-derived molecules as a therapy against inflammatory diseases.

Lymphatic Filarial Protein	Experimental Disease Model	Study Outcome	Mechanism of Action	Reference
Recombinant *B. malayi* Cystatin (rBmCys)	DSS-induced acute colitis	Down-regulated inflammatory responses and alleviated symptoms and pathology of colitis.	Elevated IL-10 + FoxP3 + Tregs, IgM + B1a cells and AAMs in the colon and peritoneal cavity. Reduced expression of Th1 and Th17 cytokines in serum and spleen.	[118,119]
rBmCys	mBSA-induced rheumatoid arthritis (RA)	Both preventive and therapeutic effects on RA. Decreased synovitis, bone erosion, fibrosis, and influx of inflammatory cells in hind paw joints.	Shift from Th1 to IL-4 and IL-10 secreting Th2 immune response.	[120,121]
Peptide fragments of rBmCys	DSS-induced acute colitis	Anti-inflammatory effect on DSS-induced colitis in mice. Reversed the gross and histopathological changes in the colitic colon.	Decreased F4/80 + TLR-4 + CD11c+ macrophages in peritoneum. Reduced LY6G+ cells and MPO+ cells and increased FoxP3 + Tregs in colon.	[122]
Recombinant *B. malayi* abundant larval transcript-2 (rBmALT-2)	DSS-induced acute colitis	More effective in preventive mode compared to therapeutic treatment against colitis.	Associated with downregulation of IFN-γ, IL-6, IL-17, and upregulation of IL-10 cytokines in spleen.	[115]
Recombinant *W. bancrofti* L-2 (rWbL2)	DSS-induced acute colitis	Reduced lymphocyte infiltration and decreased epithelial damage in colons of treated mice.	Shift towards Th2 response as reflected by increased IL-10, and decreased IFN-γ and TNF-α by splenocytes. IgG1/IgG2 ratio in the sera.	[123]
rBmALT-2, rBmCys, and rWbL-2 individually and in combinations	DSS-induced chronic colitis	All treatment strategies improved the clinicopathologic status of chronic colitis. rBmALT-2 + rBmCys showed the most prominent therapeutic effect.	Downregulated IFN-γ and TNF-α expression, upregulated IL-10, and TGF-β expression in the splenocytes. Reduction in activated NF-κB level in the colon. Increased IgG1/IgG2 ratio in the sera.	[124]
rWbL-2, rBmALT-2, and rWbL-2 + rBmALT-2	STZ-induced T1D	Led to reduced lymphocytic infiltration, islet damage, and blood glucose levels.	Decreased TNF-α and IFN-γ, and increased Il-4, IL-5, and IL-10 production in splenocytes. Elevated insulin-specific IgG1 and antigen-specific IgE antibodies in the sera.	[116]
*B. malayi* adult soluble (Bm A S) and microfilarial excretory-secretory proteins (Bm Mf ES)	STZ-induced T1D	More effective when used as curative rather than a preventive treatment. Reduced inflammatory changes in pancreatic islet cell architecture and fasting blood glucose levels.	Decreased TNF-α and IFN-γ, and increased IL-10 production in the splenocytes. Elevated anti-insulin IgG1 antibodies indicating a skewed response towards Th2 type in the sera.	[125]
*B. malayi* asparaginyl-tRNA synthetase (BmAsnRS)	T-cell transfer colitis	Resolves intestinal inflammation.	Increase in CD8^+^ T cells in the lamina propria compartment, with a corresponding increase in CD4^+^ cells in spleens of treated mice. Decrease in IFN-γ and IL-17, and increase in IL-4 and IL-10 in spleens, mesenteric lymph nodes, and lamina propria of treated mice. Induced upregulation of IL-10 and IL-22 receptors.	[126]
*Brugia malayi* K1 (BmK1)	-	Inhibits the delayed-type hypersensitivity response.	Blocked Kv1.3 receptors in human T cells. Suppressed the proliferation of rat CCR7-effector memory T cells and production of IFN-γ.	[40]

## Data Availability

Not applicable.

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
