# Peer review of "Harnessing Immune Evasion Strategy of Lymphatic Filariae: A Therapeutic Approach against Inflammatory and Infective Pathology"

_vaccines, 2022, doi:10.3390/vaccines10081235_

Round 1
Reviewer 1 Report
This review takes an unusual but innovative approach to discussion of lymphatic filariae interactions with human hosts, by looking at the potential for application of mechanisms to treat other diseases.
The English needs very, very mild editing, but overall, this is a sound, interesting manuscript.
I realize you may not know what I mean by English editing, so here is one example: "Another approach used by the parasites is called as [sic] molecular mimicry,, ....." line 135 needs to remove "as" The manuscript discusses lymphatic filariae mechanisms for avoiding host immune surveillance and attack. The authors show, in Figures 1 and 2, and well as Tables 1 and 2, how some of these mechanisms have been highjacked or co-opted to treat or vaccinate against other diseases.Author Response
We extend our sincere thanks for the encouraging words and positive criticism by the learned reviewer. We are sorry for the inconvenience. I have now edited the manuscript for language, grammar, and improved clarity. Revisions have been marked using the “Track changes” function.
Reviewer 2 Report
Priyanka Bhoj and colleagues performed a review on the mechanisms of immune evasion of the lymphatic filariae. The topic is of scientific importance as lymphatic filariasis is a neglected tropical disease that affect millions of people worldwide and new strategies of control are needed. The authors have extensively reviewed the lymphatic filariae immunomodulation strategies to evade host’s immune system. It is detailed the interaction of host immune cells with parasite virulence proteins. The authors also address immunomodulatory treatment strategies against the diseases and the host immune response generated in case of co-infection with other parasitic diseases. This last issue I considered to be of high interest, as it reflects the difficulties of the endangered populations and is less addressed in the literature. The manuscript is well written and easy to read and are there only minor points that need to be addressed before the manuscript is accepted for publication:
Line 133, 155 and 259 – please correct the location of the figure and table and captions. Figure and tables should be located where it is cited in the text.
Line 310 – visceral leishmaniasis in Southern Europe is (mainly) caused by Leishmania infantum. Please add the information.
Author Response
Dear Reviewer,
We extend our sincere thanks for the encouraging words by the learned reviewer. We appreciate the reviewer for the valuable suggestion, which we have added in the manuscript now (line no. 328). We have also placed figures, tables, and legends where they are cited in the manuscript.
Reviewer 3 Report
I am pleased with the review article description especially tables and graphical representations. Please check the reference list again as few are not describing the saying of the manucript.
Author Response
Dear Reviewer,
We thank the Reviewer for appreciating the work done by our team and letting us know this discrepancy. We are sorry for the inconvenience. We have now critically checked the reference list and corrected it.